# Nutritional quality and in vitro gas production of *Piptocoma discolor* (Kunth) Pruski forage across contrasting landscapes in the Colombian Amazon Piedmont

Faver Álvarez[1,2,3,4], Paula Andrea Ríos-Parra [2,3,4*], Fernando Casanoves[5,6], Armando Sterling[7], Isabel Cristina Molina-Botero[8]

**1** Programa de Medicina Veterinaria y Zootecnia, Facultad de Ciencias Agropecuarias, Universidad de la Amazonia, Florencia, Caquetá, Colombia, **2** Laboratorio de Evaluación de Forraje y Nutrición Animal, Centro de Investigaciones Amazónicas - CIMAZ Macagual "Cesar Augusto Estrada González, Universidad de la Amazonia, Florencia, Caquetá, Colombia, **3** Grupo de Investigación en Sistemas Agroforestales para la Amazonia -GISAPA, Universidad de la Amazonia, Florencia, Caquetá, Colombia, **4** Grupo de Investigación AMARU – Centro de Investigación, Innovación para la Sustentabilidad S.A.S. BIC, Florencia, Caquetá, Colombia, **5** CATIE-Centro Agronómico Tropical de Investigación y Enseñanza, Turrialba, Cartago, Costa Rica, **6** Programa de Doctorado en Ciencias Naturales y Desarrollo Sustentable, Facultad de Ciencias Agropecuarias, Universidad de la Amazonia, Florencia, Caquetá, Colombia, **7** Instituto de Investigación Científica Amazónica SINCHI- Programa de Biología, Facultad de Ciencias Básicas, Universidad de la Amazonia, Florencia, Caquetá, Colombia, **8** Centro Internacional de Agricultura Tropical (CIAT), Programa de Forrajes Tropicales, Palmira, Valle del Cauca, Colombia

* p.rios@udla.edu.co

**Citation:** Álvarez F, Ríos-Parra PA, Casanoves F, Sterling A, Molina-Botero IC (2026) Nutritional quality and in vitro gas production of Piptocoma discolor (Kunth) Pruski forage across contrasting landscapes in the Colombian Amazon Piedmont. PLoS One 21(3): e0345567. https://doi.org/10.1371/journal.pone.0345567

## Abstract

This study aimed to evaluate the nutritional quality and in vitro total gas production of *Piptocoma discolor* (Kunth) Pruski forage collected from two contrasting landscapes (alluvial plain and hill) in the Colombian Amazon piedmont. To achieve this, leaf and non-lignified stem samples were collected from scattered trees in each landscape and analyzed for their nutritional composition, degradability, and gas production. Crude protein (CP) levels differed significantly between landscapes ($p < 0.05$), with higher concentrations in the alluvial plain compared to the hill (14.6% vs. 9.5%, respectively). A similar trend was observed for ash, phosphorus, and nitrogen content ($p < 0.05$). However, neutral detergent fiber (NDF) content was higher in the hill landscape. The concentrations of magnesium, potassium, calcium, sodium, zinc, manganese, copper, and iron did not differ significantly between landscapes ($p < 0.05$). Regardless of the landscape, *P. discolor* exhibited moderately acceptable degradability (72%). Cumulative gas production (CGP) at 96 h was significantly higher ($p < 0.05$) for forage collected in the hill landscape compared to the alluvial plain (66 vs. 53 ml CGP/g dry matter incubated, respectively). These results confirm the potential of *P. discolor* to improve ruminant nutrition and support sustainable silvopastoral practices in the region.

**Data availability statement:** All relevant data are within the manuscript and its supplementary information files.

**Funding:** The author(s) received no specific funding for this work.

**Competing interests:** The authors have declared that there are no conflicts of interest.

## Introduction

Cattle ranching is one of the main causes of tropical forest fragmentation worldwide, altering the composition, configuration and ecological function of the biome. This transformation alters [1,2] biological cycles, biodiversity and the overall stability of ecosystems, contributing to increased climate variability [3,4]. According to Ordoñez et al. [5] the department of Caquetá, in the Colombian Amazon, is particularly affected by this problem, since several geoforms in the region have suffered important physical [6], chemical [7] and biological [8] alterations. These changes are largely due to the conversion of native forests to grasslands, which has resulted in a 19.6% decrease in soil fertility [9]. The hilly landscape, comprising 67.9% of the region, has been disproportionately affected (70% conversion) compared to the floodplain landscape (9.7%) [10]. As of 2020, the livestock population in Caquetá was approximately 2 million head of cattle [11], primarily managed within dual-purpose dairy and fattening systems. Traditionally, these production systems rely on extensive grazing over large areas of natural and improved pastures, where relatively few animals are dispersed across vast landscapes [12]. The predominant forage base consists of grasses with low nutritional value, i.e., 6.6% CP, 75% NDF and 65% digestibility, as reported in regional studies, limiting animal productivity in the Amazonian piedmont [13].

In this context, the integration of tree and shrub species into livestock diets presents a viable strategy for improving forage quality. These species offer higher crude protein content and greater biomass yield than traditional pasture grasses [14,15]. Additionally, multipurpose trees provide fruits that can substitute for grain-based feed in livestock diets [16,17], often surpassing commercial concentrates in energy content [18]. Promising species such as *Tithonia diversifolia* (Hemsl.) A. Gray*, Gmelina arborea* Roxb. ex Sm, and *Acacia mangium* Willd exhibit high biomass production, favorable nutritional profiles, and adaptability to diverse soil and climatic conditions, making them well-suited for ruminant supplementation [19]. In addition, these species contain secondary compounds such as tannins that help mitigate ammonium and methane emissions in the rumen [20] thus contributing to improved environmental sustainability in livestock production [21]. Their inclusion in forage systems promotes an efficient nutrient cycle, while adapting to the unique characteristics of Amazonian soils [22]

A particularly promising native species for the Amazon region is *P. discolor* (Kunth) Pruski [22], which thrives in secondary forests and is readily consumed by cattle grazing in pastures [23]. It should be noted that this species grows rapidly, even in degraded soils [24], tolerates high levels of precipitation typical of the Amazon region where tropical rainforest prevails [25], and is easily adaptable with a high production capacity in areas of natural regeneration, with yields of 180.56 g dry mater/plant [23], making it a valuable component of silvopastoral and agroforestry systems in the department of Guaviare, Colombia [25]. The incorporation of native trees and shrubs into livestock systems offers multiple benefits, such as increased milk production and animal weight [18], improved biodiversity conservation in agricultural landscapes, and improved resilience to climate change.

In addition, soil organisms such as earthworms and fungi improve nutrient availability and combat land degradation, supporting sustainability by linking soil health to forage quality [26]; these systems mitigate environmental impact and contribute to sustainable land management.

The wide diversity of forage species and their high nutritional value for cattle feeding in the livestock systems of the Colombian Amazon, along with their potential to reduce greenhouse gas emissions, represents an important opportunity for improving animal nutrition and mitigating the effects of climate change. These advantages contribute to enhancing both sustainability and productivity of livestock systems. However, environmental factors such as landscape type may have a direct impact on the nutritional composition of forage and the gas emissions resulting from the consumption of forage species.

In this context, hill-dominated landscapes have soils derived from highly weathered claystones and mudstones, characterized by strong acidity, high aluminum saturation, and low natural fertility due to minimal concentrations of essential nutrients such as calcium, magnesium, potassium, sodium, and phosphorus. In contrast, the soils of the alluvial plain landscapes originate from alluvial deposits, exhibit variable granulometry, and are generally shallow, poorly drained, and classified as having low to moderate natural fertility [10].

To address these gaps, we hypothesized that the nutritional characteristics of *P. discolor* forage are higher in alluvial plain landscapes, which are characterized by soils with greater natural fertility, whereas its consumption by cattle results in lower total gas production compared to hill landscapes. To test this hypothesis, this study aimed to characterize the nutritional quality and in vitro total gas production of *P. discolor* in two contrasting landscapes (alluvial plain and hill) in the Colombian Amazon piedmont.

## Materials and methods

### Study area

This study was conducted in the Municipality of Florencia, covering an area of 2,262 km². The region is situated in the piedmont between the Eastern Cordillera and the Amazon rainforest, within the Department of Caquetá [10]. The area is characterized by an average annual rainfall of 3,759 mm, an average temperature of 25.8 °C, and a relative humidity of 81%, classified as Köppen Af climate (tropical rainforest) in Colombia [24].

Two contrasting landscapes dominated by cattle grazing areas and presence of scattered *P. discolor* trees were considered [27] (i) a hill landscape with an undulating topography, featuring elevations below 300 m, broad, rounded, and elongated hilltops, and slopes ranging from 8% to 16%. The soils in this landscape are clayey, of low fertility, with variable drainage, and prone to compaction and erosion; (ii) an alluvial plain landscape, characterized by an elongated and flat topography, with slopes ranging from 0% to slightly inclined, including high and medium terraces as well as flood-prone areas. The soils in this landscape originate from alluvial-colluvial sediment deposits, exhibit moderate fertility, and have poor drainage.

The laboratory phase was carried out at the Amazonian Research Center (CIMAZ – Macagual) "Cesar Augusto Estrada González" (1°37' N, 75°36' W), located at an elevation of 300 m a.s.l., approximately 22 km south of Florencia, in the Department of Caquetá [28].

Future research could complement field-based assessments with remote sensing approaches, which offer powerful methodological tools to analyze vegetation and environmental dynamics at the landscape scale [29,30]. Integrating such approaches would provide valuable insights to expand forage studies and strengthen sustainable land-use strategies in the region.

### Sampling design and forage collection

The sample size was calculated with a 5% sampling error and a 95% confidence level using Cochran's formula [31]. Since a stratified sampling design with optimal allocation was applied, the allocation considered the variance of forage quality attributes estimated from pilot measurements (variance of 2.1 in the hill and 1.3 in the alluvial plain). This led to a greater

sampling effort in the hill stratum (36 trees) compared to the alluvial (23 trees), for a total of 59 trees. Within each land-scape unit, trees were selected based on the following criteria: (i) mature trees (8–10 years old) with a height between 16 and 20 m; (ii) trees with well-defined morphological characteristics, taking into account architecture, shape, height, and crown, without alterations due to burning or cultural practices; (iii) diameter at breast height (DBH) ≥ 10 cm; and (iv) trees free of pests and diseases.

After selection of *P. discolor* trees, leaf and non-lignified stem samples (approximately 2 kg per tree) were collected, placed in labeled paper bags, and transported to the Forage Evaluation and Animal Nutrition Laboratory at the Centro de Investigaciones CIMAZ Macagual "Cesar Augusto Estrada González."

## Nutritional composition

Laboratory samples were dried in a DHD-9030® oven (Zenith Lab, Jiangsu, China) at 105 °C for 6 h or until a constant weight was reached to determine the dry matter (DM) content, calculated as the percentage of remaining weight (AOAC, 2005; Method 950.46). Subsequently, the samples were ground using a Willey® mill (60 Hz) equipped with a 1 mm sieve. From each sample, 200 g were selected for the determination of crude protein (CP) via the Kjeldahl method (984.13) [32], acid detergent fiber (ADF), neutral detergent fiber (NDF) [33], ash content [32], and organic carbon was measured on a dry basis (Cdb) and on a wet basis (Cwb) using the Walkley and Black method [34]. The laboratory samples were dried in a DHD-9030® oven (Zenith Lab, Jiangsu, China) at 105°C for 6 hours or until a constant weight was achieved to determine the dry matter (DM) content, following the Association of Official Analytical Chemists (AOAC, 2005; Method 950.46). Subsequently, the samples were ground using a Willey® mill (60 Hz) equipped with a 1 mm sieve. From each sample, 200 g were selected for the determination of crude protein (CP) via the Kjeldahl method (984.13) [35]. Mineral quantification, including Ca, Mg, K, Na, Zn, Mn, Cu, and Fe, was performed using atomic absorption spectrophotometry with an AA7000® spectrophotometer (Shimadzu, Kyoto, Japan), while P was analyzed via UV-Vis spectrophotometry (UV 1800®, Shimadzu, Kyoto, Japan) within a wavelength range of 200–1000 nm.

## In vitro dry matter degradability

A total of 180 crucibles were selected, thoroughly washed, and placed in an oven for 12 hours. Subsequently, they were transferred to a desiccator for 5 min to equilibrate. After 96 h of incubation, the rumen kinetic residue was filtered, and the samples were dried in an oven for 24 h. Upon completion of the drying process, the samples were placed in a desiccator to reach room temperature before being weighed. The final weight of the crucible and sample was recorded to determine the *in vitro* dry matter digestibility [36,37].

## In vitro gas production

Gas production was evaluated following the protocol established by Theodorou et al. [38]. Rumen fluid was obtained from animals processed by Compañía de Ferias y Mataderos de Caquetá (COFEMA S.A.), the authorized municipal slaughterhouse in Florencia, Colombia, which operates under national animal health and welfare standards (resolution 240 of 2013).

This study did not perform any type of animal handling, experimentation or slaughter, and all samples were taken from animals that had already been slaughtered as part of routine meat processing.

The ruminal fluid was filtered through gauze and transported to the laboratory in insulated thermos flasks to maintain temperature stability. For each incubation, 0.625 g of sample was inoculated with 53.1 ml of digestion medium and 2 ml of a reducing agent in 120 ml amber glass bottles, with three replicates per treatment. Forage samples (non-lignified leaves and stems) were collected from 10 trees per landscape type, yielding a total of 180 incubation bottles (10 trees × 2 landscapes × 3 replicates × 3 ruminal fluid sources), plus 15 blank bottles (without forage). The flasks were preheated to 39 °C in a B120 digital thermostatic water bath (OVAN®, Barcelona, Spain) before adding 6.25 ml of ruminal fluid. This incubation

temperature (39 °C) was maintained throughout the experiment and readings were recorded every 3 h [38]. The total gas volume was calculated from the recorded pressure values using the following equation:

$$Y = 0.28 + 1.70x + 0.25x^2 \tag{1}$$

where $Y$ represents the gas volume produced (ml) per unit of pressure $x$

The gas production data were then fitted to the nonlinear model Gompertz [39]:

$$Y = e^{a-be^{-cx}} \tag{2}$$

where $Y$ denotes the cumulative gas production at time $x$, $a$ (> 0) represents the maximum gas production, $b$ (> 0) is the difference between initial and final gas volume at time $x$, and $c$ (> 0) describes the specific gas accumulation rate. The biological interpretation of these parameters includes: (i) TIP: Time to the inflection point (h); (ii) GIP: Gas volume at the inflection point (ml); and (iii) MRGP: Maximum rate of gas production (ml/h).

## Data analysis

To compare the means between landscape types, a t-test was conducted for each nutritional quality variable. For variables exhibiting heterogeneous variances between landscapes, Satterthwaite's correction was applied. To examine the relationships between nutritional variables, a Pearson correlation analysis was conducted [40]. The correlation matrix displaying significant relationships was visualized using the circlize package [41] in R v. 4.3.3 language [42]. Principal Component Analysis (PCA) was performed to assess associations between nutritional variables and landscapes, with results visualized using a biplot [40]. Statistical analyses related to nutritional quality were conducted using InfoStat software, v. 2019 [43].

Dry matter degradation, gas kinetics, and Gompertz model parameters were analyzed using a randomized complete block design in SAS® software, version 9.4 [44]. Each sample was incubated in triplicate across three different inoculants, with the inoculant serving as the blocking variable. Mean comparisons were performed using Tukey's test [45]. Data normality was assessed using the Shapiro-Wilk test [46] on the raw residuals within the PROC GLM procedure [47].

## Results

### Forage nutritional quality

The nutritional and mineral composition of *P. discolor* forage in two Amazonian foothill landscapes shows that the flood-plain forage showed significantly higher DM content than that of the hilly landscape (36.1% vs. 30.9%, P = 0.0458).

A similar trend was observed for CP, with *P. discolor* forage from the alluvial plain containing 1.54 times more CP than that from the hill landscape (14.6% vs. 9.5%). In contrast, NDF content was higher in the hill landscape compared to the alluvial plain (52.9% vs. 49.5%, respectively). The ADF values did not differ significantly between landscapes ($p > 0.05$).

Regarding mineral composition, ash, nitrogen, and phosphorus contents were significantly higher in the alluvial plain (9.66%, 2.34%, and 1.29%, respectively) than in the hill landscape (7.15%, 1.51%, and 0.93%, respectively), with differences ranging from 0.36% to 2.5%. However, the concentrations of Mg, K, Ca, Na, Zn, Mn, Cu, and Fe did not differ significantly between landscapes ($p > 0.05$) (Table 1).

Pearson's correlation analysis revealed the strongest significant positive correlations between CP and N, Cdb and Cwb, as well as Mg with C, Mn, and Ca (all, $r > 0.6$; $p < 0.05$) (Fig 1). In contrast, Pearson's analysis showed strong positive correlations between CP and N ($r > 0.6$, $p < 0.05$) and Mg with Ca, Mn ($r > 0.6$, $p < 0.05$), and negative correlations between NDF and DM, Cwb ($r < -0.6$, $p < 0.05$) (Fig 1).

The PCA conducted on the nutritional and mineral quality variables, considering landscape type, accounted for 43% of the total variability (Fig. 2). PC1 explained 27.5% of the variation, primarily differentiating nutritional variables such as

**Table 1. Nutritional and mineral composition (%) of *Piptocoma discolor* forage in two contrasting landscapes (hill and alluvial plain) of the Colombian Amazon piedmont. Dry matter (DM), Crude protein CP), Neutral detergent fiber (NDF), and Acid detergent fiber (ADF).**

| Content (%) | | Landscape | | | | |
|---|---|---|---|---|---|---|
| | | Hill | | Alluvial plain | | |
| | | Mean | Standard error | Mean | Standard error | p-value |
| Nutritional | DM | 30.87 | 1.78 | 36.13 | 1.78 | 0.0458 |
| | CP | 9.48 | 0.72 | 14.62 | 0.72 | <0.0001 |
| | NDF | 52.93 | 1.02 | 49.51 | 0.84 | 0.0303 |
| | ADF | 46.08 | 0.65 | 44.52 | 0.65 | 0.1031 |
| Mineral | Ash | 7.15 | 0.61 | 9.66 | 0.61 | 0.0072 |
| | Nitrogen | 1.51 | 0.11 | 2.34 | 0.11 | <0.0001 |
| | Phosphorus | 0.93 | 0.11 | 1.29 | 0.11 | 0.0310 |
| | Magnesium | 0.15 | 0.03 | 0.18 | 0.03 | 0.4212 |
| | Potassium | 1.74 | 0.23 | 1.45 | 0.18 | 0.3313 |
| | Calcium | 0.78 | 0.12 | 0.57 | 0.12 | 0.2519 |
| | Sodium | 0.01 | 3.0E-03 | 0.02 | 3.0E-03 | 0.2361 |
| | Zinc | 0.01 | 1.4E-03 | 0.004 | 1.4E-03 | 0.0820 |
| | Manganese | 0.05 | 0.01 | 0.05 | 0.01 | 0.9947 |
| | Copper | 0.01 | 2.6E-03 | 0.0045 | 2.6E-03 | 0.3560 |
| | Iron | 0.01 | 3.5E-03 | 0.01 | 3.5E-03 | 0.9145 |

CP, nitrogen, DM, carbon on a wet basis (Cwb), and carbon content on a dry basis (Cdb), which were positively associated with PC1 and showed higher values in the alluvial plain landscape. In contrast, NDF, OC, and M were more strongly associated with the hill. PC2 accounted for 17.9% of the variation, distinguishing variables such as OM, ADF, and mineral elements (Mn, Fe, Ca, K, Mg, Cu, and Na), which showed similar values across both landscapes.

## In vitro gas production

Cumulative gas production (CGP) per incubated dry matter (IDM), degradation, and their relationship are presented in Fig 2 and Table 2. At 12 h, CGP was higher in the alluvial plain landscape, reaching 24.4 ml/g IDM, while the hilly landscape yielded 19.9 ml/g IDM, 4.5 ml/g IDM less than the floodplain ($p < 0.05$). At 24 h, no significant differences were observed between landscapes ($p > 0.05$). However, at 48, 72, and 96 h post-incubation, forages from the hill landscape exhibited higher CGP values of 42, 55.9, and 65.8 ml/g IDM, respectively, compared to 35.4, 43.9, and 52.5 ml/g IDM in the alluvial plain landscape. The disparity between landscapes increased over time. On average, dry matter degradation (DMD) at 96 h was 72.1% ($p > 0.05$). CGP per degraded organic matter (DOM) was significantly higher in the hill landscape compared to the alluvial plain (102.7 vs. 78.99 ml/g).

When fitting the gas production data to the nonlinear Gompertz model, significant differences were observed ($p < 0.05$) in the cumulative gas production rate at a specific time and the maximum production rate, both of which were higher for forages from the hill landscape (Table 3). However, no significant differences ($p > 0.05$) were found between landscapes for other parameters, including maximum gas production, the difference between initial and final gas at time *x*, time at the inflection point, and gas production at the inflection point.

## Discussion

CP content (9.48% hill vs. 14.62% floodplain) was lower than 21% [21] and 20.4% [23], probably due to regrowth age and environmental factors. These results highlight the potential of this species in silvopastoral systems, due to its good growth,

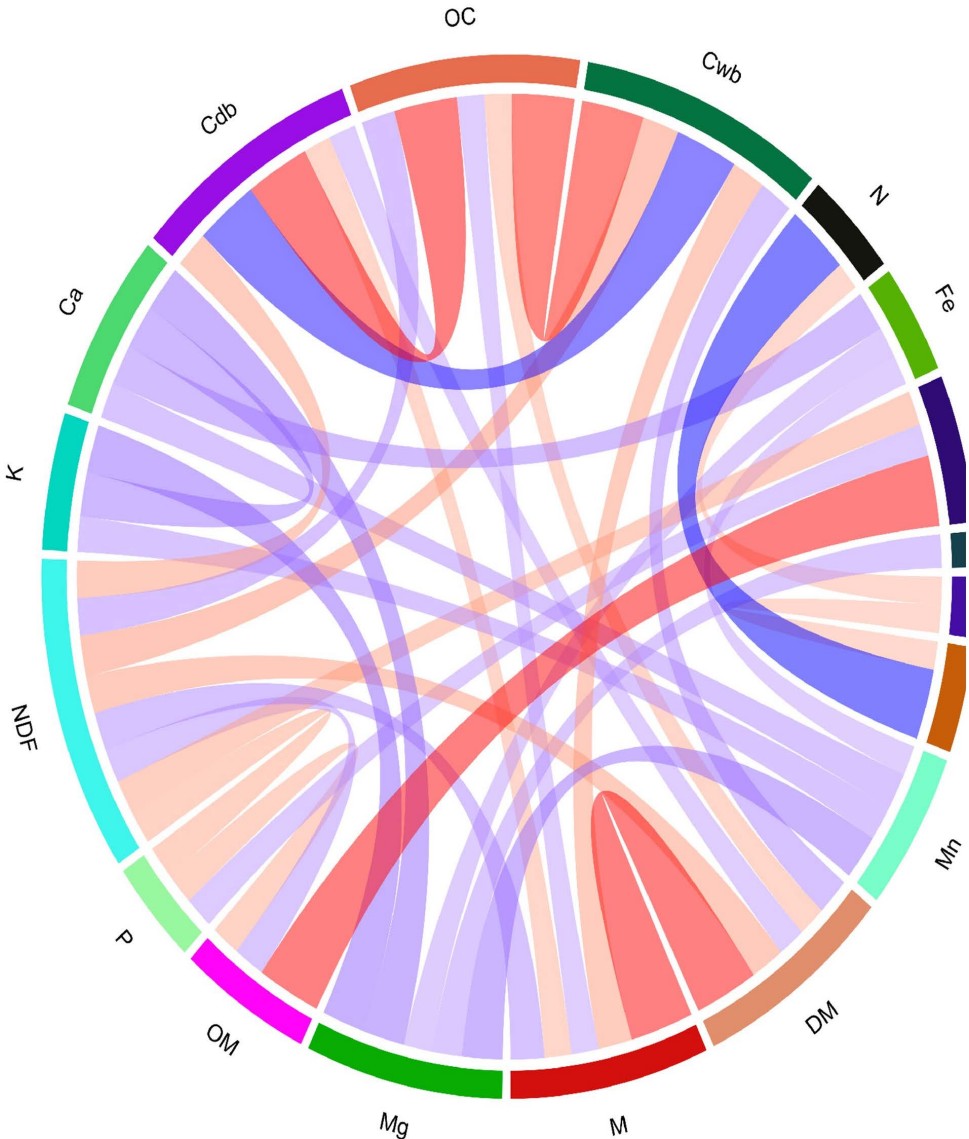

**Fig 1. Chord diagram depicting Pearson's correlation coefficients (*r*) for statistically significant relationships among the nutritional variables (p < 0.05).** Coloration and width of the ribbons indicate the direction and magnitude of the correlation, with darker blue for stronger positive (*r* > 0.4) and darker red for stronger negative (*r* < −0.3).

volume, and protein content, which exceeds that of tropical grasses, which tend to have low PC levels (≤ 7% PC) [13]. In addition, it is considered to have potential for animal feed due to its high biomass supply and nutritional composition [48] and as a promising alternative for improving the energy-protein ratio in ruminant diets.

The differences in moisture content and, consequently, DM between the hill and alluvial plain landscapes reflect the influence of landscape type on forage plants. According to Suarez et al. [49], these variations may be directly related to the species' adaptation to its environment or to potential water stress caused by soil surface compaction in the alluvial plain, leading to higher DM values in this landscape.

The NDF content was higher in samples collected from the hill than in those from the alluvial plain (52.9% vs. 49.5%, respectively). The NDF levels in this study were comparable to those reported by Riascos et al. [21] for this species

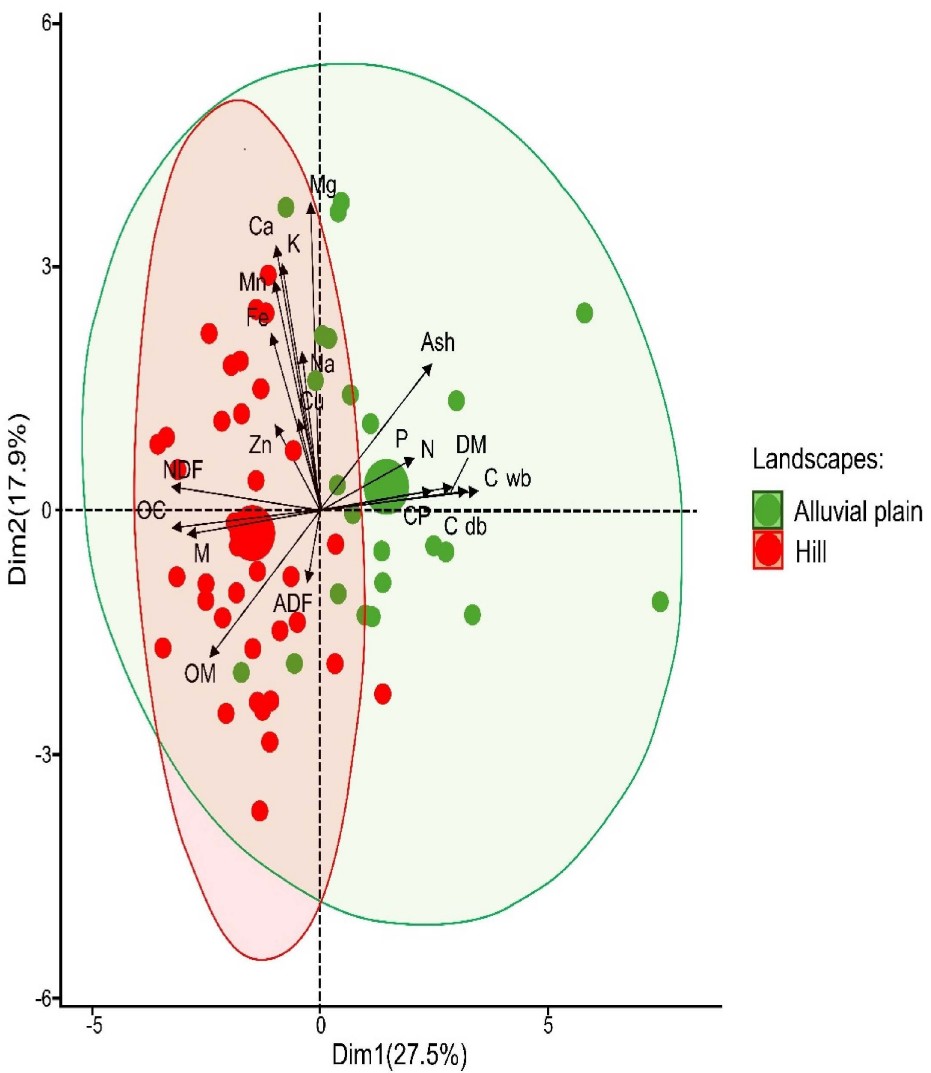

**Fig 2. Projection of the first two principal component axes, illustrating the relationships between the nutritional variables of *Piptocoma discolor* and landscape types.** Dry Matter (DM), Organic Matter (OM), carbon on a wet basis (Cwb), carbon on a dry basis (Cdb), organic carbon (CO), moisture **(M)**, Crude Protein (CP), Neutral Detergent Fiber (NDF, Acid Detergent Fiber (ADF), Nitrogen **(N)**, Phosphorus **(P)**, Magnesium (Mg), Potassium **(K)**, Calcium (Ca), Sodium (Na), Zinc (Zn), Manganese (Mn), Copper (Cu) and Iron (Fe).

**Table 2. Cumulative in vitro gas production per incubated dry matter, degraded organic matter, and dry matter degradation of *Piptocoma discolor*. MSE = mean square of the error, CGP: production of accumulated gas, IDM: incubated dry matter; DMD: dry matter degradation, DOM: degraded organic matter.**

| Landscape | CGP ml/g IDM | | | | | DMD | CGP ml/g DOM |
|---|---|---|---|---|---|---|---|
| | 12 h | 24 h | 48 h | 72 h | 96 h | (%) 96 h | 96 h |
| Hill | 19.93 | 26.87 | 41.98 | 55.89 | 65.77 | 70.72 | 102.7 |
| Alluvial plain | 24.42 | 28.17 | 35.36 | 43.93 | 52.52 | 73.60 | 78.99 |
| p-value | <0.0001 | 0.1997 | <0.0001 | <0.0001 | <0.0001 | 0.0610 | <0.0001 |
| MSE | 6.794 | 6.749 | 7.977 | 9.352 | 10.017 | 10.24 | 18.32 |

Table 3. Parameters of the Gompertz model for gas production obtained in the incubation of *Piptocoma discolor* forage in two contrasting landscapes (hill and alluvial plain) of the Colombian Amazon piedmont. MSE = Mean square error; a = Maximum gas production (ml); b = difference between initial gas and final gas at time *x*; c = Specific gas accumulation rate; TTP = Time to tipping point (h); GTP = Gas to tipping point (ml); MGPR = Maximum gas production rate (ml/h); HPI = Turning Point Time; GPI = Tipping point in gas production; TMPG = Maximum Gas Production Rate.

| Landscape | A | B | c | HPI (h) | GPI (ml) | TMPG |
|---|---|---|---|---|---|---|
| Hill | 88.11 | 0.660 | 0.020 [a] | 27.87 | 32.41 | 0.760 [a] |
| Alluvial plaine | 93.42 | 0.391 | 0.010 [b] | 34.31 | 34.36 | 0.427 [b] |
| p-value | 0.695 | 0.052 | 0.001 | 0.686 | 0.695 | 0.019 |
| CME | 14.37 | 0.078 | 0.0001 | 16.90 | 5.284 | 0.057 |

[a,b]Means in a column with different letters are statistically different (Tukey' test, *P*<0.05).

(46.60% NDF) but exceeded the values found by Maciej et al. [50] who reported NDF contents ranging from 23.4% to 36% for *Acacia decurrens* Willd. and *Sambucus nigra* L., two high-altitude forage woody species. Conversely, for lowland forage woody species, our results fall within the range (31–67% NDF) reported by Pinto-Ruiz et al. [51] for ten species (31 and 67% NDF). Regarding ADF content, the values observed in this study were 9% higher than those reported by Guayara [23] (36.1% ADF); however, ADF values as high as 27.25% have been documented in other regions, such as the Putumayo Department in Colombia. It is important to emphasize that the nutritional composition of compounds such as CP and structural carbohydrates in forage woody plants varies according to agroclimatic conditions and the number of regrowth days [52,53].

The colonization of fiber by ruminal microorganisms, their adherence to the cell wall, and the initiation of catabolic activity depend on the rumen environment (e.g., moisture, pH, temperature) and the NDF content of the diet. The degradability of forages in the rumen is closely linked to the proportion and lignification of plant cell walls [19,54].

Ash content, an indicator of total mineral content, was higher in the alluvial plain landscape than in the hill. Both values were within the range reported for other shrubs [55]. Nitrogen levels in *P. discolor* followed the same trend as ash content. Across both landscapes, mineral contents (P, N, K, Ca, Mn, Zn, Fe, Cu) were higher than those found in grasses and shrub leaves [56], suggesting that *P. discolor* could serve as a valuable mineral supplement for ruminants.

Phosphorus content in *P. discolor* was higher than that of several tropical grasses, including *Pennisetum purpureum* Schumach (0.26–0.48%), *Cenchrus ciliaris* cv. Biloela L. (0.27–0.48%), *Urochloa decumbens* (Stapf) R.D. Webster (0.19–0.22%), *Urochloa humidicola* (Rendle) Morrone & Zuloaga (0.11–0.17%), *Urochloa arrecta* (Hack. ex T. Durand & Schinz) Morrone & Zuloaga (0.14–0.22%), *Panicum maximum* Jacq. (0.25%), and *Dichantium aristatum* (Poir.) C.E.Hubb. (0.11%) [57,58]. Similarly, Ca levels were higher than those reported for tropical grasses, such as *P. maximum* (0.41%) and *D. aristatum* (0.29%) [59]. These findings are significant, as the consumption of *P. discolor* forage could help meet the mineral requirements of ruminants fed on grass-based diets. Minerals such as phosphorus and calcium are essential for ruminant health, as they play crucial roles in cellular reactions, bone and muscle formation, milk production, osmoregulation, amino acid and protein metabolism, and enzyme activity [60].

The differences in DM, ash, NDF, CP, N, and P reflect soil fertility, enhanced by biota due to the presence of earthworms and fungi, which contribute to soil health by contributing to the nutrient cycle and plant growth [26], through the degradation of organic matter, the formation of stable structures that increase soil porosity, and rapid absorption of these [61]. This continuous supply of nutrients through sediment deposition and the presence of soil fauna contrasts with the upland area, where nutrient loss is more pronounced due to leaching and suboptimal soil management, reducing the availability of N and P [62]. Laycock and Price [63] highlighted that environmental factors such as water availability, solar radiation, temperature, and soil properties significantly influence the chemical composition of forage by reducing PC content. However, Nelson and Moser [64] pointed out that each plant species has unique morphological and physiological traits that determine its adaptation, growth, and forage quality.

In this study, gas production was higher in the hill landscape than in the alluvial plain landscape, possibly due to soil characteristics influencing the nutritional composition of *P. discolor* forage. Variability in gas production and DM degradation highlights the potential of certain forages to support microbial biomass production by providing favorable fiber and protein compositions [65]. These factors enhance fermentation rates and overall forage quality [66]. Regarding CGP, *P. discolor* showed a substantial increase between 48 and 96 hours in the hill landscape. This pattern contrasts with the findings of Quintanilla et al. [67], who observed a decline in CGP over time for *Moringa oleifera* Lam. Studies suggest that forages with high cell wall content generally exhibit low digestibility and limited energy availability [68,69]. However, diets with high CP content can enhance microbial activity and improve feed intake in ruminants [70]. During the first 24 h of incubation, gas accumulation per unit of IDM did not differ between treatments. These results align with those reported by Molina-Botero et al. [59], who indicated that fermentation time influences degradation rates, thereby affecting gas production kinetics.

The DMD in *P. discolor* was higher in the alluvial plain (73.6%) than in the hill (70.7%). These values were lower than those reported by Riascos et al. [19], who found an in situ degradability of 81.5% for this species. The observed differences may stem from variations in analytical techniques. Nonetheless, *P. discolor* remains a highly degradable species, benefiting from microbial activity influenced by its fiber and CP content [71].

## Conclusions

Nutritional analysis of P. discolor from the floodplain suggests favorable properties for ruminant supplementation, with characteristics superior to those observed in the mountainous landscape. This is mainly due to its higher content of essential nutrients, such as crude protein, nitrogen, and phosphorus, along with a lower content of neutral detergent fiber, which are directly related to the production of gases in the rumen produced by the fermentation kinetics of the food consumed, the resistance time of the food in the rumen, and its degradation rate. In other words, the higher the volume of gas production, the higher the digestibility in the first few hours, demonstrating, regardless of the type of landscape, its moderate and acceptable degradation capacity.

Based on these findings, further research is recommended to determine the optimal inclusion levels of *P. discolor* in cattle diets. Such studies could assess its potential to reduce greenhouse gas emissions, thereby enhancing its overall benefits and facilitating its integration into silvopastoral systems in the Colombian Amazonian piedmont region.

Additionally, future research could incorporate landscape-scale approaches, such as remote sensing, to evaluate the adaptability of P. discolor within silvopastoral systems and expand the applicability of these findings across broader environmental contexts.

## Acknowledgments

The authors would like to thank the University of Amazonia and COFEMA S.A., especially Mr. Milton Chávez López, for their valuable support.

## Author contributions

**Conceptualization:** Paula Andrea Rios Parra, Armando Sterling.

**Data curation:** Armando Sterling, Fernando Casanoves, Isabel Cristina Molina-Botero.

**Formal analysis:** Paula Andrea Rios Parra, Armando Sterling, Fernando Casanoves, Isabel Cristina Molina-Botero.

**Investigation:** Paula Andrea Rios Parra, Faver Álvarez, Fernando Casanoves, Isabel Cristina Molina-Botero.

**Methodology:** Paula Andrea Rios Parra, Faver Álvarez, Armando Sterling, Fernando Casanoves, Isabel Cristina Molina-Botero.

**Supervision:** Faver Álvarez.

**Validation:** Faver Álvarez, Isabel Cristina Molina-Botero.

**Visualization:** Isabel Cristina Molina-Botero.

**Writing – original draft:** Paula Andrea Rios Parra, Faver Álvarez.

**Writing – review & editing:** Faver Álvarez, Armando Sterling.

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
