## [Decision Letter · Decision Letter 0]

3 Sep 2025

PONE-D-25-12452Nutritional quality and in vitro gas production of Piptocoma discolor (Kunth) Pruski forage across contrasting landscapes in the Colombian Amazon PiedmontPLOS ONE

Dear Dr. Rios Parra,

Thank you for submitting your manuscript to PLOS ONE. After careful consideration, we feel that it has merit but does not fully meet PLOS ONE’s publication criteria as it currently stands. Therefore, we invite you to submit a revised version of the manuscript that addresses the points raised during the review process.

We look forward to receiving your revised manuscript.

Kind regards,

Vitor Hugo Rodrigues Paiva, Ph.D.

Academic Editor

PLOS ONE

Journal Requirements:

3. We note that your Data Availability Statement is currently as follows:

“All relevant data are within the manuscript and its supporting information files.”

6. We note that Figure 1 in your submission contain map images which may be copyrighted. All PLOS content is published under the Creative Commons Attribution License (CC BY 4.0), which means that the manuscript, images, and Supporting Information files will be freely available online, and any third party is permitted to access, download, copy, distribute, and use these materials in any way, even commercially, with proper attribution. For these reasons, we cannot publish previously copyrighted maps or satellite images created using proprietary data, such as Google software (Google Maps, Street View, and Earth). For more information, see our copyright guidelines: http://journals.plos.org/plosone/s/licenses-and-copyright.

Additional Editor Comments:

Along with the comments and suggestions of changes raised by two experts and described below, do not forget to reply to those described on an attached annotated PDF of the manuscript

Reviewer's Responses to Questions

**Comments to the Author**

1. Is the manuscript technically sound, and do the data support the conclusions?

Reviewer #1: Yes

Reviewer #2: Yes

2. Has the statistical analysis been performed appropriately and rigorously? 

Reviewer #1: Yes

Reviewer #2: Yes

3. Have the authors made all data underlying the findings in their manuscript fully available?

Reviewer #1: Yes

Reviewer #2: Yes

4. Is the manuscript presented in an intelligible fashion and written in standard English?

Reviewer #1: Yes

Reviewer #2: Yes

5. Review Comments to the Author

Reviewer #1: This study on the nutritional quality and in vitro gas production of Piptocoma discolor across contrasting landscapes in the Colombian Amazon Piedmont provides valuable insights into its potential as a forage resource for ruminant supplementation in silvopastoral systems. Its importance lies in addressing sustainable livestock production and soil fertility challenges in a deforestation-impacted region. With the minor revisions suggested—enhancing clarity, scientific precision, and ecological context through suggested citations- the manuscript can be well-positioned for acceptance, offering a robust contribution to tropical agroforestry research.

Comments and Suggestions

1. Line 27-40 (Abstract):

Comment: The abstract is concise but could benefit from a brief mention of the study’s broader implications (e.g., sustainability or livestock productivity).

Suggestion: Add after line 40: "These results highlight the potential of P. discolor to enhance ruminant nutrition and support sustainable silvopastoral practices in the region."

2. Line 46-47 (Introduction):

Comment: The phrase "rainforest and tropical forest fragmentation" is slightly redundant since rainforests are a type of tropical forest.

Suggestion: Revise to: "Livestock production is a major driver of tropical rainforest fragmentation worldwide, altering the composition, configuration, and ecological functions of these ecosystems."

3. Line 61-62 (Introduction):

Comment: The nutritional values (6.6% CP, 75% NDF, 65% digestibility) lack a clear source or context, which might confuse readers.

Suggestion: Add qualifier: "The predominant forage base consists of grasses with low nutritional value, e.g., 6.6% CP, 75% NDF, and 65% digestibility, as reported in regional studies [13]."

4. Line 83 (Introduction):

Comment: The benefits of native trees in silvopastoral systems are mentioned, but soil biota’s role in sustainability could be emphasized.

Suggestion: After line 83, add: "Soil organisms like earthworms and fungi enhance nutrient availability and combat land degradation, supporting sustainability, cite https://doi.org/10.1002/ldr.5446 as it link soil health to forage quality, reinforcing the study’s ecological context.

5. Line 104 (Materials and Methods):

Comment: The transition to "Materials and Methods" feels abrupt without a closing statement for the Introduction.

Suggestion: Add before- Suggestion: Before line 104, add: "To address these gaps, this study evaluated the nutritional quality and rumen fermentation potential of P. discolor across two distinct landscapes."

6. Line 111 (Study Area):

Comment: "Köppen climate type A (Tropical Rainforest - Equatorial - Af)" could be simplified for readability.

Suggestion: Revise to: "classifying it as a Köppen Af (tropical rainforest) climate in Colombia [27]."

7. Line 119 (Materials and Methods - Study Area):

Comment: The study area description could suggest future research methods like remote sensing for landscape analysis.

Suggestion: After line 119, add and cite https://doi.org/10.1016/j.pce.2024.103667 , https://doi.org/10.1007/s11356-022-20305-y,
https://doi.org/10.1080/10807039.2015.1122509. "Future studies could use remote sensing, as in to map forage quality variability across Amazonian landscapes” as it offers a methodological tool for scaling up landscape-scale forage studies.

8. Line 129 (Sampling Design):

Comment: The sample size calculation lacks detail on how 36 and 23 trees were derived.

Suggestion: Add: "The sample size was calculated using Cochran’s formula [30], yielding 36 trees in the hill and 23 in the alluvial plain, totaling 59 (Fig 1)."

9. Line 147 (Nutritional Composition):

Comment: "Dry matter (DM) content" is introduced abruptly without linking to the drying process.

Suggestion: Revise to: "Samples were dried… to determine dry matter (DM) content, calculated as the percentage of weight remaining [AOAC, 2005; Method 950.46]."

10. Line 171 (In Vitro Gas Production):

Comment: "2.5 ± 0.5 years old" is vague and could be more precise.

Suggestion: Replace with: "approximately 2.5 years old (range: 2–3 years)."

11. Line 200 (Data Analysis):

Comment: "Cdb and Cwb" appear in correlation analysis (line 231) but lack prior definition.

Suggestion: Add at line 151: "Organic carbon was measured on a dry basis (Cdb) and wet basis (Cwb) using the Walkley and Black method [33]."

12. Line 214 (Results - Forage Nutritional Quality):

Comment: The sentence "The DM content was higher in the alluvial plain…" is clear but could be more engaging.

Suggestion: Revise to: "Forage from the alluvial plain showed significantly higher DM content than the hill landscape (36.1% vs. 30.9%, P = 0.0458)."

13. Line 231-234 (Results - Pearson’s Correlation):

Comment: Abbreviations "Cdb," "Cwb," "M," and "OC" are used without prior definition, potentially confusing readers.

Suggestion: Define at line 151 and simplify: "Pearson’s analysis showed strong positive correlations between CP and N (r > 0.6, P < 0.05) and Mg with Ca, Mn (r > 0.6, P < 0.05), and negative correlations between NDF and DM, Cwb (r < -0.6, P < 0.05) (Fig 2)."

14. Line 259-260 (Results - In Vitro Gas Production):

Comment: "4.5 ml/g IDM less" is awkwardly phrased.

Suggestion: Revise to: "whereas the hill landscape produced 19.9 ml/g IDM, 4.5 ml/g IDM less than the alluvial plain (P < 0.05)."

15. Line 291-292 (Discussion):

Comment: The comparison to Riascos et al. and Guayara lacks context for CP content differences.

Suggestion: Clarify: "CP content (9.48% hill vs. 14.62% alluvial plain) was lower than 21% [20] and 20.4% [23], likely due to regrowth age and environmental factors."

16. Line 335-336 (Discussion):

Comment: "Soil-type effect" is vague and could be more specific.

Suggestion: Revise to: "Differences in DM, ash, NDF, CP, N, and P reflect soil fertility, likely enhanced by soil biota like earthworms and fungi in the alluvial plain” cite https://doi.org/10.1002/ldr.5446, as it explains soil biota’s role in nutrient retention, supporting the study’s findings.

17. Line 363-364 (Conclusions):

Comment: "Bromatological analysis" may not be clear to all readers.

Suggestion: Replace with: "Nutritional analysis of P. discolor from the alluvial plain suggests favorable properties for ruminant supplementation…"

18. Line 370 (Conclusions):

Comment: The call for further research could include specific methods for broader application.

Suggestion: After line 370, add: "Remote sensing, as in [Bhattacharjee et al., 2024], could assess landscape-scale adaptability of P. discolor in silvopastoral systems." https://doi.org/10.1016/j.pce.2024.103667 provides a method to expand the study’s scope, enhancing future research.

19. Line 374-378 (Acknowledgments):

Comment: The acknowledgment is brief and could be more formal.

Suggestion: Revise to: "The authors thank the Universidad de la Amazonia and COFEMA S.A., especially Mr. Milton Chávez López, for their valuable support."

Reviewer #2: The paper, titled "Nutritional quality and in vitro gas production of Piptocoma discolor (Kunth) Pruski forage across contrasting landscapes in the Colombian Amazon Piedmont", addresses an important and timely topic. I found the subject matter of the article fascinating and read the manuscript with great interest. The paper aligns well with the scope of the journal.

Main Questions and Relevance

The main question this research tackles is how the nutritional quality and in vitro total gas production of Piptocoma discolor forage differs when collected from two very distinct landscapes—alluvial plain and hill—in the Colombian Amazon piedmont. They're basically trying to see if where this plant grows affects how good it is as feed for ruminants and how much gas (like methane, which is a big deal for climate change) it produces.

I definitely consider this topic original and highly relevant to the field of animal nutrition and sustainable agriculture, especially in tropical regions. It directly addresses a pretty significant gap: while we know a lot about common forage species, there's less research on the potential of native trees and shrubs, particularly how their nutritional value might vary based on local environmental conditions like soil type. This is crucial for developing sustainable silvopastoral systems. It’s not just about finding new feed, but understanding how to use existing, native resources more effectively in specific ecological contexts. Many studies just look at one area, but here, they’ve thought about the different landscapes.

Contribution to the Subject Area

What this paper adds, really, is a specific and detailed look at Piptocoma discolor in a region where such data is pretty scarce. Compared to other published material that might focus on more common forage species or broad comparisons, this study provides empirical data on a promising native species, highlighting how landscape-level differences can impact key nutritional parameters like crude protein, fiber, and mineral content, as well as gas production. The finding that P. discolor's degradability is moderately acceptable regardless of landscape is quite useful, and the differences in cumulative gas production are certainly noteworthy. It really helps to fill in the picture for those looking to implement sustainable livestock practices in the Colombian Amazon.

However, I believe that in its current form, it has several shortcomings.

Specific comments

Introduction

The introduction does a good job of setting the stage, emphasizing the importance of silvopastoral systems in the Amazonian context. However, it feels like it could be a tad more concise in certain parts. For example, some of the initial sentences on deforestation and climate change, while true, are quite broad and could be streamlined to get to the core problem statement a bit faster. Also, when discussing previous work on P. discolor, perhaps a slightly more critical look at why this study is needed, beyond just stating its promising nature, would be beneficial. What specific questions remain unanswered by earlier research that this paper uniquely addresses? The hypothesis is clearly stated, which is a big plus, but a stronger link between the literature review and the novelty of their specific landscape comparison would be good.

In the Introduction, line 45, it says "fragmentation worldwide, altering the composition, configuration, and ecological functions of these ecosystems." The phrase "these ecosystems" is a bit vague, could perhaps specify "rainforest and tropical forest ecosystems" again for clarity.

Still in the Introduction, line 55 talks about the "hilly landscape, which dominates the region (67.9%) has been disproportionately affected (70%)". The percentages here are very close but refer to different things; maybe a quick rephrasing for precision would help avoid a misreading, like "the hilly landscape, comprising 67.9% of the region, has been disproportionately affected (70% of the conversion)."

In the Introduction, when discussing previous work on P. discolor and its promising nature, there might be room to add more citations if there's research that specifically details why this plant is so "promising." It’s kinda generally stated.

In line 73, the secondary compounds must be listed, maybe they are tannins, please see andi cite: 10.1016/j.vas.2025.100434 to support your statement.

Materials and Methods

The methodology is generally well-described, but I have a few specific points that need attention. First, on the sampling design: While the sample size calculation is mentioned as per Cochran [30], it would be helpful to briefly explain how that calculation was applied, perhaps noting the variance estimates used, especially given the differing number of trees sampled in each landscape (36 vs. 23). This just adds to the transparency. Second, when describing the rumen fluid collection, mentioning that cattle had "free access to fresh water" is good, but any information on their feed prior to slaughter would be immensely valuable. The diet of the donor animals can significantly influence the rumen microbial population, which, in turn, impacts in vitro gas production results. This is a pretty important detail for replicability and interpretation. Finally, for the gas production calculations, Equation (1) is provided, but it's not clear what "x" represents in that specific equation. Is it pressure? Is it time? The text says "per unit of pressure x," which suggests pressure, but clarification would remove any ambiguity. It just seems a little bit confusing for the reader.

In the Materials and Methods, line 107, there's a typo in "Municipality of Florencia (Fig 1), covering an area of 107 2,262 km2." The "107" seems like a stray line number or something that got included by mistake.

Also in Methods, line 131, it mentions "height ranging from 16 to 20 m". Is this referring to the height of the trees or the sampling height on the trees? I'm assuming trees, but clarifying would be good.

Supporting your in vitro methods I suggest citing 10.1080/1828051X.2021.1899063.

The authors should consider including references to support the statistical methods used in the analysis. Some key areas to address include: 10.3390/ani13061107 for pearson correlation; 10.3389/fvets.2024.1332207 for PCA; 10.29261/pakvetj/2020.067 for Shapiro wilk; 10.1080/01652176.2024.2347928 for tukey; 10.1186/s12917-023-03823-w for GLM.

Results

The results section is largely clear, but the presentation could be improved, particularly in the text versus table consistency. For instance, in Table 1, the "I" for Iron seems like a typo and should probably be "Fe" for consistency with other mineral abbreviations and standard chemical symbols. Just a small thing, but those sorts of details can distract a reader. When discussing the PCA, Figure 3 is very illustrative, but the explanation in the text about PC1 and PC2 could be slightly more explicit about which variables are loading strongly in which direction along those axes, beyond just listing them. For example, stating that "higher carbon content on a dry basis (Cdb) observed in the alluvial plain landscape" is a bit vague – is Cdb associated with the alluvial plain on PC1? Making that more explicit helps the reader interpret the biplot. Also, the description of Figure 2's chord diagram notes "Coloration and width of the ribbons indicate the direction and magnitude of the correlation, with blue representing positive values (r > 0.4 ) and red representing negative values (r < -0.3)." This is a bit of a broad range for "strong" correlations. Clarifying what the bands represent within that range (e.g., darker blue for stronger positive, etc.) would be helpful for full interpretation.

In Results, line 215, "1.54 times more CP" is clear, but "P < 0.05" is typically presented in tables. If it's in the text, usually it’s fine, but just be consistent whether you always mention p-values in the text for significant differences or only in the tables.

Table 1 has "Iron (I)" which should be "Iron (Fe)" for standard chemical notation. Just a little correction.

Discussion

The discussion does a commendable job of comparing the findings with existing literature, which is a major strength. However, sometimes the connections between the study's specific findings and the broader implications could be drawn more strongly. For example, while the differing CP and NDF levels are discussed in comparison to other studies, there could be more in-depth speculation on why these differences occur beyond just soil type – perhaps linking it to specific plant physiology responses to nutrient availability in those contrasting landscapes. There's a slight disconnect when you say "Differences in DM, ash, NDF, CP, nitrogen, and phosphorus content indicate a soil-type effect on forage quality," then directly after, "This continuous supply of nutrients through sediment deposition contrasts with the highland zone, where nutrient loss is more pronounced due to leaching and suboptimal soil management." This is good, but it could be woven more deeply into the prior comparisons of CP and NDF. Also, the statement that "CGP per degraded organic matter (DOM) was significantly higher in the hill landscape compared to the alluvial plain (102.7 vs. 78.99 ml/g)" is a very interesting result, but the discussion doesn't fully delve into the implications of this difference. Why might the hill landscape forage produce more gas per unit of degraded organic matter? This is a key finding that warrants more detailed interpretation in the discussion.

In the Discussion, line 290, "The CP content observed in P. discolor trees in this study (9.48% in the hill landscape vs. 14.62% in the alluvial plain) was lower than the values reported by Riascos et al. [20], who found 21% CP in trees with 60-day regrowth, and by Guayara [23] who reported 20.4% CP in trees with 45-day regrowth." There's a slight awkwardness in flow here, perhaps "The CP content observed in P. discolor trees in this study (9.48% vs. 14.62% for hill and alluvial plain landscapes, respectively) was lower than..." might read a bit better.

Generally, throughout the text, ensure that species names like Piptocoma discolor are consistently italicized wherever they appear. I think they mostly are, but it's always worth a double-check.

More concretely, in the Discussion section, around line 310, where it talks about ADF values as high as "27.25% have been documented in other regions, such as the Putumayo Department in Colombia." This specific claim about Putumayo needs a citation. It sounds like a direct piece of data from another study, and if it's not cited, it could look like unbacked information, which is a big no-no.

Conclusions

The conclusions are consistent with the evidence presented and largely address the main question posed. They effectively summarize the key findings regarding nutritional properties and degradability. The statement that P. discolor "possesses favorable nutritional properties for ruminant supplementation" is well-supported. However, the conclusion that it's "directly associated with gas production at the rumen level" could be expanded slightly to clarify how this association is demonstrated by their data – i.e., whether higher nutrient content led to lower or higher gas production, linking back to the differing CGP results. It’s a bit vague at the end. Perhaps also, reiterate the implications for climate change, given it's mentioned in the introduction.

References

The references appear mostly appropriate and cover relevant literature. The formatting seems consistent with the journal's requirements. No major issues noted here, but I didn't do a deep dive into every single one. Just make sure all linked URLs are still active.

Tables and Figures

Overall, the tables and figures are clear and help in understanding the data. Table 1 is very comprehensive, but as noted, the "I" for Iron should probably be "Fe". A quick check on units is always good too, just to be extra sure everything is clear. Figure 1 is a fantastic addition, clearly showing the sampling locations, which is vital for understanding the study's geographical context. Figure 2, the chord diagram, is visually appealing and effectively conveys the correlations, though the clarification on the interpretation of ribbon width/color within the defined ranges would be good. Figure 3, the PCA biplot, is also very well done and effectively visualizes the relationships between variables and landscapes. My only thought there is, perhaps a slightly larger font for the variable labels within the plot would improve readability for those of us with older eyes!

Figure legends: For Figure 3, in the legend for the PCA, "CO" is listed for organic carbon, but in the Methods section, it was "OC." Consistency here would be good, sticking to one abbreviation. Just a very small thing.

6. PLOS authors have the option to publish the peer review history of their article (what does this mean? ). If published, this will include your full peer review and any attached files.

**Do you want your identity to be public for this peer review?** For information about this choice, including consent withdrawal, please see our Privacy Policy .

Reviewer #1: No

Reviewer #2: No

---

## [Author Response · Author response to Decision Letter 1]

8 Jan 2026

Florencia, Colombia, October 1, 2025

Editors-in-Chief

PLOS ONE

Dear Editors

Please find enclosed a revised version of our manuscript: “Nutritional quality and in vitro gas production of Piptocoma discolor (Kunth) Pruski forage across contrasting landscapes in the Colombian Amazon Piedmont”.

We sincerely thank the editor and reviewers for their encouraging, positive evaluation and their advice on improving the manuscript before its acceptance for publication. All were reasonable and thoroughly considered when revising our manuscript. We hope that this revised version of the manuscript is improved with the modifications and can now be considered for publication.

EDITOR

Additional requirements.

Response: For the purposes of the investigation, access was granted to a slaughterhouse in the city of Florencia Caquetá, with the direct authorization of the manager and person in charge of the establishment, after explaining the intentions and interest behind the visit.

3. We note that your Data Availability Statement is currently as follows:

“All relevant data are within the manuscript and its supporting information files.”

Response: Thank you for making the publication of the article feasible, for this process, the article contains the data that were used in the study.

6. We note that Figure 1 in your submission contain map images which may be copyrighted. All PLOS content is published under the Creative Commons Attribution License (CC BY 4.0), which means that the manuscript, images, and Supporting Information files will be freely available online, and any third party is permitted to access, download, copy, distribute, and use these materials in any way, even commercially, with proper attribution. For these reasons, we cannot publish previously copyrighted maps or satellite images created using proprietary data, such as Google software (Google Maps, Street View, and Earth). For more information, see our copyright guidelines: http://journals.plos.org/plosone/s/licenses-and-copyright.

Additional Editor Comments:

Along with the comments and suggestions of changes raised by two experts and described below, do not forget to reply to those described on an attached annotated PDF of the manuscript

Response: All the editor’s comments were taken into account and addressed in accordance with their instructions and the journal’s guidelines.

REVIEWER #1

1. Lines 27-40 (Abstract):

Comment: The abstract is concise but could benefit from a brief mention of the broader implications of the study (e.g., sustainability or livestock productivity).

• Suggestion: Add after line 40 “These results highlight the potential of P. discolor to improve ruminant nutrition and support sustainable silvopastoral practices in the region.”

Response: Lines 40-41. These results confirm the potential of P. discolor to improve ruminant nutrition and support sustainable silvopastoral practices in the region.

1. Lines 46-47 (Introduction):

• Comment: The phrase “rainforest and fragmentation of tropical forests” is slightly redundant since rainforests are a type of tropical forest.

• Suggestion: Revise to: “Livestock production is one of the main drivers of tropical rainforest fragmentation worldwide, altering the composition, configuration, and ecological functions of these ecosystems.”

Response: Lines 48-49. Cattle ranching is one of the main causes of tropical forest fragmentation worldwide, altering the composition, configuration and ecological function of the biome.

1. Lines 61-62 (Introduction):

• Comment: The nutritional values (6.6% CP, 75% NDF, 65% digestibility) lack a clear source or context, which could confuse readers.

• Suggestion: Add qualifier: “The predominant forage base consists of pastures with low nutritional value, for example, 6.6% CP, 75% NDF, and 65% digestibility, as reported in regional studies [13].”

Response: Lines 60-62. The document was modified, as mentioned by the reviewer.

1. Line 83 (Introduction):

• Comment: The benefits of native trees in silvopastoral systems are mentioned, but the role of soil biota in sustainability could be emphasized.

• Suggestion: After line 83, add: "Soil organisms such as earthworms and fungi improve nutrient availability and combat soil degradation, supporting sustainability. Cite https://doi.org/10.1002/ldr.5446 as it links soil health to forage quality, reinforcing the ecological context of the study.

Response: Line 85-87. In addition, soil organisms such as earthworms and fungi improve nutrient availability and combat land degradation, supporting sustainability by linking soil health to forage quality [64], these systems mitigate environmental impact and contribute to sustainable land management.

1. Line 104 (Materials and Methods):

• Comment: The transition to “Materials and Methods” seems abrupt without a closing statement for the Introduction.

• Suggestion: Add before—Suggestion: Before line 104, add: “To address these gaps, this study evaluated the nutritional quality and ruminal fermentation potential of P. discolor in two distinct landscapes.”

Response: Line 101. We add this text: “To address these gaps,” before hypothesis for closing statement for the Introduction.

1. Line 111 (Study area):

• Comment: “Köppen climate type A (Tropical Rainforest - Equatorial - Af)” could be simplified to make it easier to read.

• Suggestion: Revise to: “classifying it as Köppen climate Af (tropical rainforest) in Colombia [27].”

Response: Line 113-114. classified as Köppen Af climate (tropical rainforest) in Colombia.

Line 119 (Materials and methods - Study area):

• Comment: The description of the study area could suggest future research methods such as remote sensing for landscape analysis.

• Suggestion: After line 119, add and cite https://doi.org/10.1016/j.pce.2024.103667,
https://doi.org/10.1007/s11356-022-20305-y,
https://doi.org/10.1080/10807039.2015.1122509. “Future studies could use remote sensing, as in to map forage quality variability across Amazonian landscapes” as it offers a methodological tool to expand forage studies at the landscape scale.

Response: Lines 126-129. We appreciate your valuable suggestion. Following your recommendation, we incorporated a statement highlighting the potential of remote sensing as a methodological tool to complement field-based assessments and expand forage studies at the landscape scale. This addition was included after line 119 in the revised manuscript. To maintain coherence with the scope of our study, we selected and cited the two references most closely related to our research context: https://doi.org/10.1016/j.pce.2024.103667,
https://doi.org/10.1007/s11356-022-20305-y

1. Line 129 (Sampling design):

• Comment: The sample size calculation lacks details on how 36 and 23 trees were obtained.

• Suggestion: Add: "The sample size was calculated using Cochran's formula [30], giving 36 trees on the hill and 23 on the floodplain, for a total of 59 (Fig. 1).

Response: Lines 132-137. This change was made: The sample size was calculated using Cochran's formula [32], giving 36 trees on the hill and 23 on the floodplain, for a total of 59 (Fig 1).

1. Line 147 (Nutritional composition):

• Comment: “Dry matter (DM) content” is introduced abruptly without linking to the drying process.

• Suggestion: Revise to: “The samples were dried... to determine the dry matter (DM) content, calculated as the percentage of remaining weight [AOAC, 2005; Method 950.46].”

Response: Lines 151-153. Laboratory samples were dried in a DHD-9030® oven (Zenith Lab, Jiangsu, China) at 105°C for 6 hours or until a constant weight was reached to determine the dry matter (DM) content, calculated as the percentage of remaining weight.

1. Line 171 (In vitro gas production):

• Comment: “2.5 ± 0.5 years” is vague and could be more precise.

• Suggestion: Replace with: “approximately 2.5 years (range: 2-3 years)”.

Response: Lines 178-180. This change was made: Rumen fluid was collected at the time of slaughter from three healthy cattle approximately 2.5 years old (range: 2- 3 years) at the municipal slaughterhouse of Florencia.

1. Line 200 (Data analysis):

• Comment: “Cdb and Cwb” appear in the correlation analysis (line 231) but are not defined beforehand.

Organic carbon was measured on a dry basis (Cdb) and on a wet basis (Cwb) using the Walkley and Black method.

• Suggestion: Add to line 151: “Organic carbon was measured on a dry basis (Cdb) and on a wet basis (Cwb) using the Walkley and Black method [33].”

Response: The definitions of Cdb and Cwb were revised, placing the previous definitions in lines 157 Nutritional Composition.

2. Line 214 (Results - Nutritional quality of forage):

• Comment: The sentence “The DM content was higher in the floodplain...” is clear, but could be more appealing.

• Suggestion: Revise to: “The floodplain forage showed significantly higher DM content than that of the hilly landscape (36.1% vs. 30.9%, P = 0.0458).”

Response: This change was made: Line 223-224. The floodplain forage showed significantly higher DM content than that of the hilly landscape (36.1% vs. 30.9%, P = 0.0458)."

2. Lines 231-234 (Results - Pearson Correlation):

Observation: The abbreviations “Cdb,” “Cwb,” “M,” and “OC” are used without prior definition in , which may confuse readers.

• Suggestion: Define in line 151 and simplify: “Pearson's analysis showed strong positive correlations between CP and N (r > 0.6, P < 0.05) and Mg with Ca, Mn (r > 0.6, P < 0.05), and negative correlations between FDN and MS, Cwb (r < -0.6, P < 0.05) (Fig 2).”

Response: Line 241-243. This change was made: Pearson's analysis showed strong positive correlations between CP and N (r > 0

---

## [Decision Letter · Decision Letter 1]

26 Feb 2026

PONE-D-25-12452R1Nutritional quality and in vitro gas production of Piptocoma discolor (Kunth) Pruski forage across contrasting landscapes in the Colombian Amazon PiedmontPLOS One

Dear Dr. Rios Parra,

Thank you for submitting your manuscript to PLOS ONE. After careful consideration, we feel that it has merit but does not fully meet PLOS ONE’s publication criteria as it currently stands. Therefore, we invite you to submit a revised version of the manuscript that addresses the points raised during the review process.

We look forward to receiving your revised manuscript.

Kind regards,

Vitor Hugo Rodrigues Paiva, Ph.D.

Academic Editor

PLOS One

Journal Requirements:

Reviewers' comments:

Reviewer's Responses to Questions

**Comments to the Author**

1. If the authors have adequately addressed your comments raised in a previous round of review and you feel that this manuscript is now acceptable for publication, you may indicate that here to bypass the “Comments to the Author” section, enter your conflict of interest statement in the “Confidential to Editor” section, and submit your "Accept" recommendation.

Reviewer #1: (No Response)

Reviewer #2: All comments have been addressed

2. Is the manuscript technically sound, and do the data support the conclusions?

Reviewer #1: Yes

Reviewer #2: Yes

3. Has the statistical analysis been performed appropriately and rigorously? 

Reviewer #1: Yes

Reviewer #2: Yes

4. Have the authors made all data underlying the findings in their manuscript fully available?

Reviewer #1: Yes

Reviewer #2: Yes

5. Is the manuscript presented in an intelligible fashion and written in standard English?

Reviewer #1: Yes

Reviewer #2: Yes

6. Review Comments to the Author

Reviewer #1:**The manuscript is scientifically sound and significantly improved. If the authors confirm the inclusion of the minimal raw data set and add the specific map attribution, it is ready for publication .**

Reviewer #2: well done! no additional comments from my side, all the points were properly addressed by the authors

7. PLOS authors have the option to publish the peer review history of their article (what does this mean? ). If published, this will include your full peer review and any attached files.

**Do you want your identity to be public for this peer review?** For information about this choice, including consent withdrawal, please see our Privacy Policy .

Reviewer #1: **Yes:** SADASHIV CHATURVEDI

Reviewer #2: No

---

## [Author Response · Author response to Decision Letter 2]

5 Mar 2026

Florencia, Colombia, March 3, 2026

Editors-in-Chief

PLOS ONE

Dear Editors:

We attach a revised version of our manuscript: “Nutritional quality and in vitro gas production of forage Piptocoma discolor (Kunth) Pruski in contrasting landscapes of the Colombian Amazonian piedmont”.

We sincerely thank the editor for his positive and encouraging evaluation and advice to improve the manuscript before acceptance for publication. All comments were reasonable and were taken into account when revising our manuscript. We hope that this revised version of the manuscript has improved with the modifications and can now be considered for publication.

EDITOR

Additional requirements.

1. If the authors have adequately addressed your comments raised in a previous round of review and you feel that this manuscript is now acceptable for publication, you may indicate that here to bypass the “Comments to the Author” section, enter your conflict of interest statement in the “Confidential to Editor” section, and submit your "Accept" recommendation.

Reviewer #1: (No Response)

Reviewer #2: All comments have been addressed

Response: All comments mentioned by the reviewers have been taken into account for the improvement of the article, its submission, and its acceptance.

2. Review Comments to the Author Please use the space provided to explain your answers to the questions above. You may also include additional comments for the author, including concerns about dual publication, research ethics, or publication ethics. (Please upload your review as an attachment if it exceeds 20,000 characters)

Reviewer #1: The manuscript is scientifically sound and significantly improved. If the authors confirm the inclusion of the minimal raw data set and add the specific map attribution, it is ready for publication .

Reviewer #2: well, done! no additional comments from my side, all the points were properly addressed by the authors

Response: Given the rejection of the map's inclusion in the manuscript, the decision was made to remove it, as a previous version had been sent with justification and corrections but was rejected. Therefore, the map is no longer part of the manuscript. The responses sent to the journal editor for review are listed below, but they were not accepted.

The maps, basemaps, shapefiles, and all other cartographic inputs used in the figure were obtained from the Open Data portal of the Agustín Codazzi Geographic Institute (IGAC). All geographic layers—including municipal and departmental boundaries, drainage networks, roads, and other spatial elements—originate exclusively from this official platform, which is publicly available for download and free use under its open data license.

The map was produced using QGIS, an open-source geographic information system that enables the processing, analysis, and visualization of spatial data. All cartographic layers obtained from IGAC’s Open Data repository were integrated, processed, and rendered within this software environment to generate the final figure.

---

## [Editor Report · Decision Letter 2]

9 Mar 2026

Nutritional quality and in vitro gas production of Piptocoma discolor (Kunth) Pruski forage across contrasting landscapes in the Colombian Amazon Piedmont

PONE-D-25-12452R2

Dear Dr. Rios Parra,

We’re pleased to inform you that your manuscript has been judged scientifically suitable for publication and will be formally accepted for publication once it meets all outstanding technical requirements.

Kind regards,

Vitor Hugo Rodrigues Paiva, Ph.D.

Academic Editor

PLOS One
---

## [Editor Report · Acceptance letter]

PONE-D-25-12452R2

PLOS One

Dear Dr. Rios Parra,

I'm pleased to inform you that your manuscript has been deemed suitable for publication in PLOS One. Congratulations! Your manuscript is now being handed over to our production team.

Kind regards,

on behalf of

Dr. Vitor Hugo Rodrigues Paiva

Academic Editor

PLOS One